**DOI: 10.1038/s41467-017-01819-3**　　**OPEN**

# A glimpse of gluons through deeply virtual compton scattering on the proton

M. Defurne et al.[#]

The internal structure of nucleons (protons and neutrons) remains one of the greatest outstanding problems in modern nuclear physics. By scattering high-energy electrons off a proton we are able to resolve its fundamental constituents and probe their momenta and positions. Here we investigate the dynamics of quarks and gluons inside nucleons using deeply virtual Compton scattering (DVCS)—a highly virtual photon scatters off the proton, which subsequently radiates a photon. DVCS interferes with the Bethe-Heitler (BH) process, where the photon is emitted by the electron rather than the proton. We report herein the full determination of the BH-DVCS interference by exploiting the distinct energy dependences of the DVCS and BH amplitudes. In the regime where the scattering is expected to occur off a single quark, measurements show an intriguing sensitivity to gluons, the carriers of the strong interaction.

#A full list of authors and their affliations appears at the end of the paper

The dynamics of quarks and gluons inside the nucleon are governed by the strong interaction, described by the theory of quantum chromodynamics. At a distance scale of the nucleon radius, perturbative computations cannot be performed because of the large value of the strong coupling constant $\alpha_S$. To unravel the internal dynamics of the nucleon and answer fundamental questions from the origin of its spin to the mechanism of confinement, lepton-scattering experiments have proven to be a powerful tool. Indeed, elastic scattering allows to access the transverse spatial distribution of charge and current in the nucleon through measurements of its electric and magnetic form factors, whereas parton distribution functions measured in deep inelastic experiments provide information on the longitudinal momentum carried by the confined quarks and gluons. Developed in the mid-90s, the generalized parton distributions (GPDs)[1–3] provide a higher level of information and encode correlations between the transverse position and the longitudinal momentum of quarks and gluons inside the nucleon[4]. Being a $\frac{1}{2}$-spin particle, the nucleon is described by four chiral-even GPDs $\left\{H, E, \tilde{H}, \tilde{E}\right\}$ and their chiral-odd counterparts more commonly called transversity GPDs, for each quark flavor and for gluons[5].

GPDs are accessible through deep virtual photoproduction processes: a high-energy virtual photon scatters off the proton and all the subsequent particles in the final state are identified[6–15].

**Fig. 1** A few examples of DVCS diagrams. At leading-order in perturbative quantum chromodynamics (QCD) (**a**), the virtual photon with four-momentum $q$ interacts with a single quark (single straight line) from the proton $p$, in the limit $Q^2 = -q^2$ much larger than the proton mass squared. Subsequently, the active quark emits a real photon with four-momentum $q'$. The recoil proton has four-momentum $p'$. Perturbation theory can be used to calculate the part of the amplitude above the (dashed) factorization line, whereas GPDs encode the non-perturbative structure of the nucleon. At next-to-leading order in perturbative QCD (**b**), a gluon (curly line) from the proton splits into a quark-antiquark pair and the quark absorbs the virtual photon. **c** An example of deeply virtual Compton scattering (DVCS) diagram at next-to-leading twist illustrating a quark-gluon correlation. The average longitudinal momentum fraction carried by the active parton (quark/gluon) is $x$ and $-2\xi$ is the longitudinal momentum transfer. The helicity of the photons contributing to the leading-twist amplitudes are specified in parenthesis

The high-energy scale introduced by the virtual photon (or equivalently its short distance resolution scale) ensures that the reaction is governed by perturbative dynamics of quarks and gluons ($\alpha_S \ll 1$). In this work we focus on the process in which a single high-energy photon is emitted by the scattered proton, the so-called deeply virtual Compton scattering (DVCS). Similarly to holography, measuring not only the magnitude but also the phase of the DVCS amplitude allows to perform three-dimensional (3D) images of the proton internal structure. The phase is accessible through the quantum-mechanical interference of DVCS with the Bethe-Heitler (BH) process, where the photon is emitted by the electron rather than the proton.

We report accurate measurements of the photon electro-production cross section. Our results show an unexpected sensitivity to the gluon content of the proton. In addition, by using different incident beam energies, we were able to isolate the contribution of the BH-DVCS interference term from the pure DVCS$^2$ amplitude.

## Results

**The DVCS amplitude.** Collinear factorization theorems[16,17] demonstrate that at sufficiently high energy, the DVCS amplitude is a convolution of a perturbative kernel with the GPDs of the nucleon—which describe the nucleon's non-perturbative structure (Fig. 1). These convolutions, called Compton form factors (CFFs), can be classified according to photon-helicity states. With $\lambda$ and $\lambda'$ the helicity state of the virtual photon and outgoing real photon, respectively, we distinguish three kinds of photon-helicity-dependent CFFs $\mathcal{F}_{\lambda\lambda'} \in \left\{\mathcal{H}_{\lambda\lambda'}, \mathcal{E}_{\lambda\lambda'}, \tilde{\mathcal{H}}_{\lambda\lambda'}, \tilde{\mathcal{E}}_{\lambda\lambda'}\right\}$[18]: the helicity-conserved CFFs ($\mathcal{F}_{++}$), which describe diagrams for which the virtual and the outgoing photons have the same helicity state, the transverse-to-transverse helicity-flip CFFs ($\mathcal{F}_{-+}$) for which the virtual and the outgoing photons have opposite helicities, and the longitudinal-to-transverse helicity-flip CFFs ($\mathcal{F}_{0+}$) describing the contribution of a longitudinally polarized virtual photon. The CFFs are also classified according to the inverse power of $Q \equiv \sqrt{Q^2}$ with which they enter the DVCS amplitude. This power is called the twist, and is equal to the dimension minus the spin of the corresponding operator. The leading-twist (LT) CFFs are $\mathcal{F}_{++}$ and $\mathcal{F}_{-+}$, which are twist-2. CFFs $\mathcal{F}_{0+}$ are twist-3, i.e., $\frac{1}{Q}$-suppressed with respect to the LT CFFs. Note that the gluon contribution (Fig. 1 top right) while twist-2, is suppressed by a factor of $\alpha_S$ (next-to-leading-order; NLO).

To experimentally study DVCS, the virtual photon in the initial state is produced via the scattering of a multi-GeV electron off a proton. Consequently, DVCS events have an electron and a proton ($ep$) in the initial state, with a final state composed of the scattered electron, the recoil proton and the high-energy photon ($ep\gamma$). However, the final photon of the reaction $ep \rightarrow ep\gamma$ can also be emitted by either the incoming or scattered electron instead of the proton, a competing channel called BH. Therefore, the exclusive photon electroproduction $ep \rightarrow ep\gamma$ cross section of a polarized electron beam of energy $k$ off an unpolarized target of mass $M$ (Fig. 2) can be written as[19]:

$$\frac{d^4\sigma(h)}{dQ^2 dx_B dt d\phi} = \frac{d^2\sigma_0}{dQ^2 dx_B} \times \left[\left|\mathcal{T}^{\mathrm{BH}}\right|^2 + \left|\mathcal{T}^{\mathrm{DVCS}}(h)\right|^2 - \mathcal{I}(h)\right] \quad (1)$$

where $\phi$ is the angle between the leptonic and hadronic planes defined by the Trento convention[20], $h$ the lepton helicity, and $\mathcal{I}$ is the interference of the virtual Compton $\mathcal{T}^{\mathrm{DVCS}}$ and BH $\mathcal{T}^{\mathrm{BH}}$ amplitudes.

The interference between BH and DVCS provides a way to independently access the real and imaginary parts of CFFs. At leading order (LO), the imaginary part of $\mathcal{F}_{++}$ is directly related

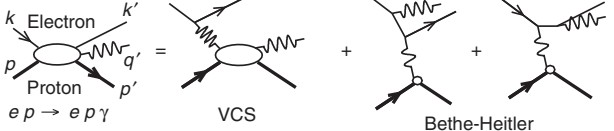

**Fig. 2** Lowest-order diagrams for $ep \rightarrow ep\gamma$. The momentum four-vectors of external particles are labeled on the left. The net four-momentum transfer to the proton is $\Delta_\mu = (q - q')_\mu = (p' - p)_\mu$. In the virtual Compton scattering (VCS) amplitude, the (spacelike) virtuality of the incident photon is $Q^2 = -q^2 = -(k - k')^2$. The Bjorken variable $x_B$ is defined as $x_B = Q^2/(2q \cdot P)$. In the Bethe-Heitler amplitude, the virtuality of the incident photon is $-\Delta^2 = -t$.

**Table 1 Kinematic settings experiment E07–007**

| $Q^2$ (GeV$^2$) | $x_B$ | $E^{beam}$ (GeV) | $-t$ (GeV$^2$) |
|---|---|---|---|
| 1.50 | 0.36 | 3.355 | 0.18, 0.24, 0.30 |
|  |  | 5.55 |  |
| 1.75 | 0.36 | 4.455 | 0.18, 0.24, 0.30, 0.36 |
|  |  | 5.55 |  |
| 2.00 | 0.36 | 4.455 | 0.18, 0.24, 0.30, 0.36 |
|  |  | 5.55 |  |

Three $Q^2$ settings were measured at constant value of $x_B$ and at two different incident beam energy $E^{beam}$. The values of $-t$ at which cross sections were determined are reported in the last column of the table

to the corresponding GPD at $x = \xi$:

$$\mathcal{R}e\,\mathcal{F}_{++} = \mathcal{P}\int_{-1}^{1} dx \left[\frac{1}{x-\xi} - \kappa\frac{1}{x+\xi}\right] F(x, \xi, t),$$
$$\mathcal{I}m\,\mathcal{F}_{++} = -\pi[F(\xi, \xi, t) + \kappa F(-\xi, \xi, t)], \quad (2)$$

where $\kappa = -1$ if $F \in \{H, E\}$ and 1 if $F \in \{\tilde{H}, \tilde{E}\}$. Recent phenomenology uses the LT and LO approximation in order to extract or parametrize GPDs, which translates into neglecting $\mathcal{F}_{0+}$ and $\mathcal{F}_{-+}$ and using the relations of Eq. (2)[21–23].

**The experiment**. We report herein measurements of helicity-dependent and helicity-independent photon electroproduction cross sections with high statistical accuracy in Hall A of Jefferson Lab. The H($\vec{e}, e'\gamma$)p cross section was measured at $x_B = 0.36$ for three $Q^2$-settings. Data for each $Q^2$-value were taken with two incident beam energies and binned in $-t$. Kinematics are summarized in Table 1. The present experimental study was initiated to separate the DVCS-BH interference and DVCS$^2$ contributions to the $ep \rightarrow ep\gamma$ cross section, by exploiting the different energy dependences of the BH and DVCS amplitudes. Until now, only the asymmetry between incident electron and positron beams could be used to constrain the real part of this interference[9,24].

In experiment E07–007 a longitudinally polarized electron beam impinged on a 15-cm-long liquid H$_2$ target. Beam polarization was continuously measured by the Hall A Compton polarimeter and found to be $72 \pm 2\%_{sys}$ on average. Scattered electrons were detected in the left high-resolution spectrometer (HRS)[25]. Events were triggered by the coincidence of a scintillator plane (S2m) and a signal in a gas Čerenkov counter. The HRS $\delta p/p \sim 10^{-4}$ momentum resolution and $\delta\theta \sim 0.6$ mr horizontal angular resolution provide a precise measurement of the electron kinematics and interaction vertex. Tracking efficiency was known to 0.5%. The final state photon was detected in an electromagnetic calorimeter consisting on an $16 \times 13$ array of PbF$_2$ crystals. Its energy resolution was measured to be 2.4% at 4.2 GeV, with ~3 mm spatial resolution.

The exclusivity of the reaction is ensured by a cut on the $ep \rightarrow e\gamma X$ missing mass squared $M_X^2 = (k + p - k' - q')^2$ (Fig. 3). The number of events $N_C$ below the missing mass cut $M_C^2$ is the sum of four contributions:

$$N_C = N_{ep \rightarrow ep\gamma} + N_{\pi^0 - 1\gamma} + N_{acc} + N_{SIDIS}, \quad (3)$$

with $N_{ep \rightarrow ep\gamma}$ the number of exclusive photon events, $N_{\pi^0 - 1\gamma}$ the contamination from $\pi^0$ decays that yield only one photon in the calorimeter, $N_{acc}$ the number of electron-photon accidental coincidences, and $N_{SIDIS}$ the contamination from semi-inclusive events $ep \rightarrow ep\gamma X$. The contamination caused by asymmetric $\pi^0$ decays with respect to the pion momentum was estimated by simulating thousands of decays for each $\pi^0$ identified in the data and computing the likelihood for each to yield only one photon in the experimental acceptance. The subtraction of $N_{acc}$ was performed by analyzing events where the scattered electron and the detected photon were not in coincidence. In addition, we applied a 800 MeV energy cut on the photon to remove most of the accidental background and required a value of $M_X^2 > 0.5$ GeV$^2$ to increase the signal/background ratio (Fig. 3). We also applied a $M_X^2 < 1$ GeV$^2$ cut so that $N_{SIDIS}$ is < 1% of exclusive $N_{ep \rightarrow ep\gamma}$ events. The significant fraction of exclusive photon events with a missing mass squared higher than $M_C^2$ is corrected by applying the same cut to the Monte-Carlo simulation used to compute the experimental acceptance. This fraction of events removed varies from bin to bin since the width and position of the exclusive signal may slightly change from one bin to another. The compact experimental setup provides a very flat geometrical acceptance, except at the edges of the detectors, where it drops to 75–30% depending on the kinematic setting. The energy resolution of the calorimeter was smeared locally in order to match the missing mass resolution observed in the experimental data, and the point-to-point systematic uncertainty associated to the exclusivity cut estimated to be 2%.

The Monte-Carlo simulation is based on the GEANT4 toolkit and includes real and virtual radiative corrections following the procedure described in ref. [12] and based on calculations by Vanderhaeghen et al.[26]. A 2% point-to-point systematic uncertainty has been attributed to the radiative corrections and a 1% correlated uncertainty to the HRS acceptance model[27]. The simulation is used to account for bin migration effects in $t$ and $\phi$ (around 10% in average) due to detector resolution and Bremsstrahlung radiation[12], with 1% point-to-point systematic uncertainty. An additional bin in $t$ is used to correct for bin migration in and out of the largest $|t|$ – bin. We also include 2% correlated uncertainty for the integrated luminosity and data acquisition dead-time correction and 0.5% for trigger efficiency, which yields a total systematic uncertainty of 3.9% for the unpolarized cross sections and 4.4% for the helicity-dependent cross sections. The total systematic uncertainties are comparable to the typical values of the statistical uncertainties on the unpolarized cross sections.

**Photon electroproduction cross sections**. The scattering amplitude is a Lorentz invariant quantity, but the deeply virtual scattering process nonetheless defines a preferred axis (light-cone axis) for describing the scattering process. At finite $Q^2$ and non-zero $t$, there is an ambiguity in defining this axis, though all definitions converge as $Q^2 \rightarrow \infty$ at fixed $t$. Belitsky et al.[18] decompose the DVCS amplitude in terms of photon-helicity states where the light-cone axis is defined in the plane of the four-vectors $q$ and $P$. This leads to the CFFs defined previously. Recently, Braun et al.[28] proposed an alternative decomposition, which defines the light-cone axis in the plane formed by $q$ and $q'$, resulting in a new set of CFFs $\mathbb{F}_{\lambda\lambda'}$ parameterizing the DVCS

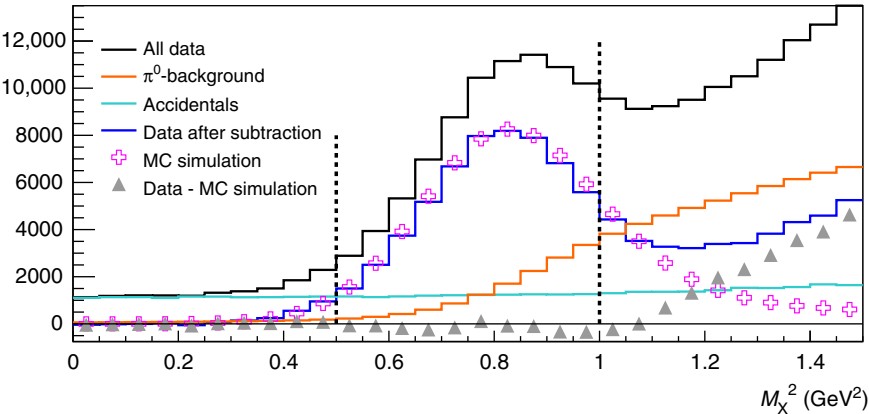

**Fig. 3** Missing mass squared distribution. The black histogram presents the raw data. Accidental and $\pi^0$ backgrounds are shown in green and orange, respectively. The subtraction of the accidental and $\pi^0$ contributions from the raw data is displayed in blue. The Monte-Carlo simulation is represented by the open crosses, whereas the triangles show the estimated inclusive yield obtained by subtracting the simulation from the background-subtracted data. The vertical dotted lines illustrate the two cuts applied on $M_X^2$ in the analysis. This figure corresponds to the kinematic setting $E^{beam} = 4.455$ GeV and $Q^2 = 1.75$ GeV$^2$, integrated over $t$

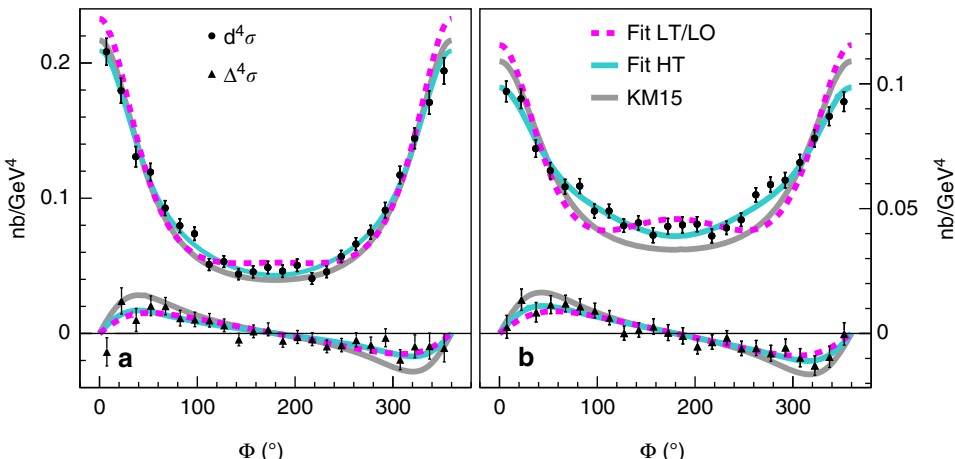

**Fig. 4** Beam helicity-dependent and helicity-independent cross sections. Unpolarized cross sections are represented with black circles and polarized cross sections with black triangles. The kinematic setting shown corresponds to $Q^2 = 1.75$ GeV$^2$, $x_B = 0.36$, and $t = -0.30$ GeV$^2$. The beam energies are $E^{beam} = 4.455$ GeV (**a**) and $E^{beam} = 5.55$ GeV (**b**). Bars show s.d. statistical uncertainties, calculated as the squared root of the number of detected events and propagated to the measured cross sections. Dashed lines represent the result of the LT/LO fit with $\mathbb{H}_{++}$, $\mathbb{E}_{++}$, $\tilde{\mathbb{H}}_{++}$, and $\tilde{\mathbb{E}}_{++}$. Solid lines show the result of the HT fit with $\mathbb{H}_{++}$, $\tilde{\mathbb{H}}_{++}$, $\mathbb{H}_{0+}$, and $\tilde{\mathbb{H}}_{0+}$. Curves for the NLO fit ($\mathbb{H}_{++}$, $\tilde{\mathbb{H}}_{++}$, $\mathbb{H}_{-+}$, and $\tilde{\mathbb{H}}_{-+}$) overlap with the HT fit and are not shown. Results of the KM15[29] fit to previously published DVCS data are also presented

amplitude, and argue that this is more convenient to account for kinematical power corrections of $\mathcal{O}(t/Q^2)$ and $\mathcal{O}(M^2/Q^2)$. The bulk of these corrections can be included by rewriting the CFFs $\mathcal{F}_{\lambda\lambda'}$ in terms of $\mathbb{F}_{\lambda\lambda'}$ using the following map[28]:

$$\mathcal{F}_{++} = \mathbb{F}_{++} + \frac{\chi}{2}[\mathbb{F}_{++} + \mathbb{F}_{-+}] - \chi_0 \mathbb{F}_{0+}, \qquad (4)$$

$$\mathcal{F}_{-+} = \mathbb{F}_{-+} + \frac{\chi}{2}[\mathbb{F}_{++} + \mathbb{F}_{-+}] - \chi_0 \mathbb{F}_{0+}, \qquad (5)$$

$$\mathcal{F}_{0+} = -(1+\chi)\mathbb{F}_{0+} + \chi_0[\mathbb{F}_{++} + \mathbb{F}_{-+}], \qquad (6)$$

where kinematic parameters $\chi_0$ and $\chi$ are defined as follows

(Eq. 48 of ref.[28]):

$$\chi_0 = \frac{\sqrt{2}Q\tilde{K}}{\sqrt{1+\epsilon^2}(Q^2+t)} \propto \frac{\sqrt{t_{\min}-t}}{Q}, \qquad (7)$$

$$\chi = \frac{Q^2 - t + 2x_B t}{\sqrt{1+\epsilon^2}(Q^2+t)} - 1 \propto \frac{t_{\min}-t}{Q^2}. \qquad (8)$$

Within the $\mathbb{F}_{\mu\nu}$-parameterization, the LT and LO approximation consists in keeping $\mathbb{F}_{++}$ and neglecting both $\mathbb{F}_{0+}$ and $\mathbb{F}_{-+}$. Nevertheless, as a consequence of Eqs. (5) and (6), $\mathcal{F}_{0+}$ and $\mathcal{F}_{-+}$ are no longer equal to zero since proportional to $\mathbb{F}_{++}$. The functions that can be extracted from data to describe the 3D

structure of the nucleon become:

$$\mathcal{F}_{++} = \left(1 + \frac{\chi}{2}\right)\mathbb{F}_{++}, \mathcal{F}_{0+} = \chi_0 \mathbb{F}_{++}, \mathcal{F}_{-+} = \frac{\chi}{2}\mathbb{F}_{++}. \quad (9)$$

A numerical application gives $\chi_0 = 0.25$ and $\chi = 0.06$ for $Q^2 = 2\,\mathrm{GeV}^2$, $x_B = 0.36$, and $t = -0.24\,\mathrm{GeV}^2$. Considering the large size of the parameters $\chi_0$ and $\chi$, these kinematical power corrections cannot be neglected in precision DVCS phenomenology, in particular in order to separate the DVCS-BH interference and DVCS$^2$ contributions. Indeed, when the beam energy changes, not only do the contributions of the DVCS-BH interference and DVCS$^2$ terms change but also the polarization of the virtual photon changes, thereby modifying the weight of the different helicity amplitudes.

Figure 4 presents the beam helicity-dependent and helicity-independent cross sections measured in one kinematic bin, at two different values of the incident beam energy. The KM15[29] line shows a LT and LO global fit without kinematically suppressed

**Table 2 Results of the cross-section fits**

| Fit description | LO/LT | Higher twist | NLO |
|---|---|---|---|
| **Helicity states** | **++** | **++/0+** | **++/−+** |
| $t = -0.18\,\mathrm{GeV}^2$ | 250 | 204 | 206 |
| $t = -0.24\,\mathrm{GeV}^2$ | 367 | 206 | 208 |
| $t = -0.30\,\mathrm{GeV}^2$ | 415 | 189 | 190 |

Values of $\chi^2$ (ndf = 208) obtained in the leading-order, leading-twist (++); higher-twist (++/0+); and next-to-leading-order (++/−+) scenarios. The fit is not performed at the highest value of −t because of the lack of full acceptance in $\phi$, resulting in a large statistical uncertainty. The fits include statistical and point-to-point systematic uncertainties

power corrections, which was able to reproduce all currently available DVCS data, from collider to fixed-target experiments.

**Discussion**

Neglecting the (logarithmic) $Q^2$-evolution of the CFFs between 1.5 and $2\,\mathrm{GeV}^2$, we have performed a combined fit of all our data at constant $x_B$ and $t$. For each −$t$ bin, this fit includes the helicity-dependent and helicity-independent cross sections at two values of beam energy and all three values of $Q^2$. Point-to-point systematic uncertainties (3.2% total) were added to statistical uncertainties quadratically when performing the fit. The effect of correlated systematic uncertainties was found negligeable.

The LO/LT fit is shown in Fig. 4 for $t = -0.30\,\mathrm{GeV}^2$, in which the free parameters are the real and imaginary parts of $\mathbb{H}_{++}$, $\tilde{\mathbb{H}}_{++}$, $\mathbb{E}_{++}$, and $\tilde{\mathbb{E}}_{++}$. This fit reproduces very poorly the angular distribution of the data yielding a value of $\chi^2/\mathrm{ndf} = 415/208$. Indeed, the strong enhancement of the $\cos\phi$-harmonics in the DVCS$^2$ amplitude originated by the large size of $\chi_0$ translates into the bump in the dashed line around $\phi = 180°$ for $E^{\mathrm{beam}} = 5.550\,\mathrm{GeV}$. Two additional fits were performed including either $\{\mathbb{H}_{0+}, \tilde{\mathbb{H}}_{0+}\}$ to include genuine twist-3 contributions or $\{\mathbb{H}_{-+}, \tilde{\mathbb{H}}_{-+}\}$ to include gluon-transversity GPD contributions arising at NLO . In both of these latter fits $\mathbb{E}_{++}$ and $\tilde{\mathbb{E}}_{++}$, expected to have the smallest contributions, were set to zero, thus keeping constant the number of free parameters. Including $\mathbb{E}_{++}$ and $\tilde{\mathbb{E}}_{++}$ yields similarly good fits, though. The fit to the data is much better ($\chi^2/\mathrm{ndf} = 190/208$) for both the higher-twist (HT) or the NLO scenarios than for the LO/LT case. This conclusion also holds for the lower −$t$ bins, as summarized in Table 2. We observe the crucial role of gluons in the description of the process, either through the quark-gluon correlations involved in HT diagrams or

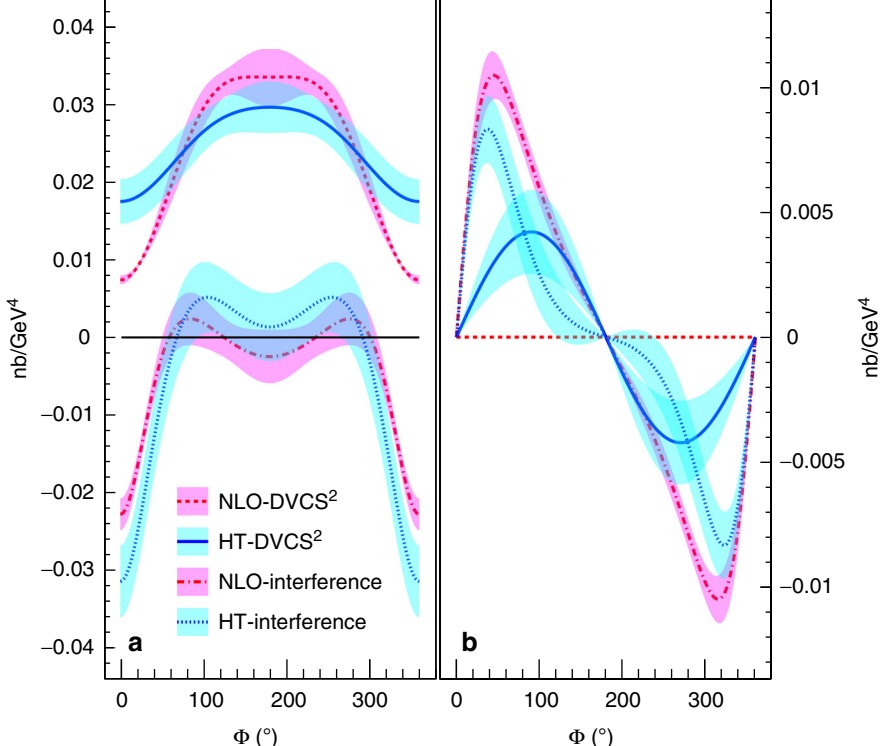

**Fig. 5** A generalized Rosenbluth separation. DVCS$^2$ and DVCS-BH interference contributions are shown at $Q^2 = 1.75\,\mathrm{GeV}^2$, $x_B = 0.36$, $t = -0.30\,\mathrm{GeV}^2$, and $E^{\mathrm{beam}} = 5.55\,\mathrm{GeV}$ for the helicity-independent (**a**) and helicity-dependent (**b**) cross sections. Solid and dotted lines represent these contributions for the twist-3 (HT) scenario; dashed and dashed-dotted lines correspond to the NLO scenario. Bands show s.d. statistical uncertainties. A DVCS$^2$ contribution appears in the helicity-dependent cross section only if there is a contribution from the longitudinal polarization of the virtual photon (HT scenario)

through next-to-leading effects implying gluon-transversity GPDs. This pioneer analysis including the kinematical power corrections recently calculated for DVCS demonstrate that the LT approximation is no longer sufficient to describe the accuracy of these new data.

Within both successful fit scenarios, the DVCS$^2$ and the BH-DVCS interference terms are well separated, as presented in Fig. 5: we denote this procedure a generalized Rosenbluth separation. In particular, we note a significant DVCS$^2$ contribution in the HT scenario to the helicity-dependent cross section, assumed to be a purely interference term in DVCS phenomenology up to now. In addition, the real part of the BH-DVCS interference (helicity-independent cross section) is extracted in these kinematics.

In conclusion, we measured beam helicity-dependent and helicity-independent photon electroproduction cross sections off a proton target for three $Q^2$-values ranging from 1.5 to 2 GeV$^2$ at $x_B = 0.36$. Each kinematic setting was measured at two incident beam energies. Using this data set, we demonstrated the sensitivity of high-precision DVCS data to twist-3 and/or higher-order contributions through a phenomenological study including kinematical power corrections. Within either a pure HT or a pure NLO scenario, both legitimate at our moderate values of $Q^2$, a statistically significant experimental separation of the DVCS$^2$ and DVCS-BH interference terms is achieved. While HT effects in GPD models[30,31] are of the order of magnitude observed, no numerical estimate of NLO contributions due to gluon-transversity GPDs are available. Advances in global analyses can include these next-order contributions, rich with information about parton correlations inside the nucleon[32,33]. Finally, a new program has started at Jefferson Lab to measure deep virtual exclusive scattering with electron beams up to 11 GeV. For a given $x_B$, the reach in $Q^2$ will span at least a factor of two. This broader reach provides the potential to discriminate between the two scenarios (HT or NLO), as the cross sections in the two scenarios (for the same GPDs) have different energy and $Q^2$ dependencies at fixed $x_B$.

**Data availability**. Data that support the findings of this study are available in HEPData with the identifier http://dx.doi.org/10.17182/hepdata.78261.

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

## Acknowledgements

We thank V. Braun, M. Diehl, H. Moutarde, and K. Kumerički for valuable discussions about the phenomenological aspects of these results. We acknowledge essential work of the Jefferson Lab accelerator staff and the Hall A technical staff. This work was supported by the Department of Energy (DOE), the National Science Foundation, the French Centre National de la Recherche Scientifique, the Agence Nationale de la Recherche, the Commissariat à l'énergie atomique et aux énergies alternatives, and P2IO Laboratory of Excellence. Jefferson Science Associates, LLC, operates Jefferson Lab for the U.S. DOE under U.S. DOE contract DE-AC05-060R23177.

## Author contributions

M.D., A.M.J., M.B., P.B., C.D., E.F., C.E.H., M.M., C.M.C., R.P., M.N.H.R., J.R. and F.S. participated to the data analysis. Z.A., H.A., K.A., K.A., V.B., M.Br., W.B., A.C., M.C., S.C., C.C., J.P.C., C.W. de J., R. d. L., C.D., A.D., L.E.F., R.E., D.F., M.F., E.F., S.F., F.G., D.G., A.G., O.G., S.G., J.G., O.H., D.H., T.H., T.H., J.H., M.H., C.E.H., S.I., F.I., H.K., A. K., C.K., S.K., I.K., J.J.L., R.L., E.L., M.M., J.M., D.J.M., P.M., M.M., F.M., D.M., R.M., M. M., C.M.C., P.N.T., N.N., A.P., V.P., Y.Q., A.R., M.N.H.R., S.R., J.R., G.R., K.S., A.S., B.S.,

L.S., A.S., S.S., P.S., M.L.S., R.S., V.S., C.S., W.A.T., G.M.U., D.W., B.W., H.Y., Z.Y., X.Z., J.Z., B.Z., Z.Z., X.Z. and P.Z. contributed to the data taking. All authors discussed the results and collectively wrote the paper.

## Additional information

**Competing interests:** The authors declare no competing financial interests.

M. Defurne[1], A. Martí Jiménez-Argüello[2,3], Z. Ahmed[4], H. Albataineh[5], K. Allada[6], K.A. Aniol[7], V. Bellini[8], M. Benali[9], W. Boeglin[10], P. Bertin[9,11], M. Brossard[9], A. Camsonne[11], M. Canan[12], S. Chandavar[13], C. Chen[14], J.-P. Chen[11], C.W. de Jager[11], R. de Leo[15], C. Desnault[2], A. Deur[11], L. El Fassi[16], R. Ent[11], D. Flay[17], M. Friend[18], E. Fuchey[1,9,19], S. Frullani[20], F. Garibaldi[20], D. Gaskell[11], A. Giusa[8], O. Glamazdin[21], S. Golge[22], J. Gomez[11], O. Hansen[11], D. Higinbotham[11], T. Holmstrom[23], T. Horn[24], J. Huang[6], M. Huang[25], C.E. Hyde[9,12], S. Iqbal[7], F. Itard[9], H. Kang[26], A. Kelleher[27], C. Keppel[11], S. Koirala[12], I. Korover[28], J.J. LeRose[11], R. Lindgren[29], E. Long[30], M. Magne[9], J. Mammei[31], D.J. Margaziotis[7], P. Markowitz[10], M. Mazouz[32], F. Meddi[20], D. Meekins[11], R. Michaels[11], M. Mihovilovic[33], C. Muñoz Camacho[2,9], P. Nadel-Turonski[11], N. Nuruzzaman[14], R. Paremuzyan[2], A. Puckett[34], V. Punjabi[35], Y. Qiang[11], A. Rakhman[4], M.N.H. Rashad[12], S. Riordan[36], J. Roche[13], G. Russo[8], F. Sabatié[1], K. Saenboonruang[29,37], A. Saha[11], B. Sawatzky[11,17], L. Selvy[30], A. Shahinyan[38], S. Sirca[33], P. Solvignon[11,39], M.L. Sperduto[8], R. Subedi[40], V. Sulkosky[6], C. Sutera[8], W.A. Tobias[29], G.M. Urciuoli[41], D. Wang[29], B. Wojtsekhowski[11], H. Yao[17], Z. Ye[29], X. Zhan[42], J. Zhang[11], B. Zhao[27], Z. Zhao[29], X. Zheng[29] & P. Zhu[29]

[1]Irfu, CEA, Université Paris-Saclay, 91191 Gif-sur-Yvette, France. [2]Institut de Physique Nucléaire CNRS-IN2P3, 15 rue Georges Clémenceau, 91406 Orsay, France. [3]Facultad de Física Universidad de Valencia, Carrer del Dr. Moliner 50, 46100 Burjassot, Spain. [4]Syracuse University, 900 South Crouse Ave., Syracuse, NY 13244, USA. [5]Texas A&M University-Kingsville, Engineering Complex, 700 University Blvd, Kingsville, TX 78363, USA. [6]Massachusetts Institute of Technology, 77 Massachusetts Ave, Cambridge, MA 02139, USA. [7]California State University, 5151 State University Dr, Los Angeles, CA 90032, USA. [8]INFN/Sezione di Catania, Via S. Sofia, 62, 95125 Catania, Italy. [9]Clermont université, université Blaise Pascal, CNRS/IN2P3, 4 Avenue Blaise Pascal, 63178 Aubire Cedex, France. [10]Florida International University, 11200 SW 8th St, Miami, FL 33199, USA. [11]Thomas Jefferson National Accelerator Facility, 12000 Jefferson Ave, Newport News, VA 23606, USA. [12]Old Dominion University, 5115 Hampton Blvd, Norfolk, VA 23529, USA. [13]Ohio University, 123 University Terrace, 1 Ohio University, Athens, OH 45701, USA. [14]Hampton University, 100 E Queen St, Hampton, VA 23668, USA. [15]Università di Bari, Piazza Umberto I, 1, 70121 Bari, Italy. [16]Rutgers The State University of New Jersey, 7 College Ave, New Brunswick, NJ 08901, USA. [17]Temple University, 1801 N Broad St, Philadelphia, PA 19122, USA. [18]Carnegie Mellon University, 5000 Forbes Ave, Pittsburgh, PA 15213, USA. [19]University of Connecticut, 2390 Alumni Drive, Unit 3206, Storrs, CT 06269, USA. [20]INFN/Sezione Sanità, Viale Regina Elena 299, 00161 Roma, Italy. [21]Kharkov Institute of Physics and Technology, Akademichna St, 1, Kharkov, Kharkiv Oblast 61000, Ukraine. [22]North Carolina Central University, 1801 Fayetteville St, Durham, NC 27707, USA. [23]Longwood University, 201 High St, Farmville, VA 23909, USA. [24]The Catholic University of America, 620 Michigan Ave NE, Washington, DC 20064, USA. [25]Duke University, Physics Bldg., Science Dr., Campus Box 90305, Durham, NC 27708, USA. [26]Seoul National University, 1 Gwanak-ro, Gwanak-gu, Seol, South Korea. [27]College of William and Mary, Department of Physics, P.O. Box 8795, Williamsburg, VA 23187, USA. [28]Tel Aviv University, P.O. Box 39040, Tel Aviv 6997801, Israel. [29]University of Virginia, 382 McCormick Rd, Charlottesville, VA 22904, USA. [30]Kent State University, 800 E Summit St, Kent, OH 44240, USA. [31]University of Massachusetts, 1126 Lederle Graduate Research Tower (LGRT), Amherst, MA 01003, USA. [32]Faculté des Sciences de Monastir, Avenue de l'environnement, 5019 Monastir, Tunisia. [33]University of Ljubljana, Kongresni trg 12, 1000 Ljubljana, Slovenia. [34]Los Alamos National Laboratory, Los Alamos, NM 87545, USA. [35]Norfolk State University, 700 Park Avenue, Norfolk, VA 23504, USA. [36]Stony Brook University, 100 Nicolls Rd, Stony Brook, NY 11794, USA. [37]Kasetsart University, 50 Thanon Ngam Wong Wan, Khwaeng Lat Yao, Khet Chatuchak, Krung Thep, Maha Nakhon 10900, Thailand. [38]Yerevan Physics Institute, 2. Alikhanian Br. Street, Yerevan 0036, Armenia. [39]University of New Hampshire, 105 Main St, Durham, NH 03824, USA. [40]George Washington University, 2121 I St NW, Washington, DC 20052, USA. [41]INFN/Sezione di Roma, Piazzale Aldo Moro 2, 00185 Roma, Italy. [42]Argonne National Laboratory, 9700 Cass Ave, Lemont, IL 60439, USA. C.W. de Jager, S. Frullani, A. Saha and P. Solvignon are deceased

