## [Peer Review File · Nature Communications]

Reviewers' comments:

Reviewer #1 (Remarks to the Author):

The manuscript presents a new measurement of the scattering cross section for the reaction $e+p \rightarrow e+p+\gamma$.

The scattering amplitude for this process receives two types of contributions: photon radiation off the electron (called Bethe-Heitler process, BH) and photon radiation off the proton's constituents (called deeply virtual compton scattering, DVCS). The BH contribution can be predicted in QED, while the DVCS contribution can in principle be used to extract information on the inner structure of the proton. Both reactions interfere in the squared amplitude that determines the cross section.

The measurement presented here improves upon earlier measurements of this scattering reaction in terms of precision and kinematical reconstruction. Moreover, fits to various phenomenological models are provided. These fits, combined with the energy dependence of the cross section, are used subsequently to disentangle the cross section contributions from the square of DVCS and the DVCS-BH interference. This (model dependent) extraction, Fig.5, is the main novel result of this paper.

Although scientifically sound and interesting to its specific subfield as a proof-of-principle on how to disentangle the different contributions to the $e+p \rightarrow e+p+\gamma$ reaction, this result falls very much short off the promise of the paper that is made in the title and abstract: providing a 3D holographic measurement of the proton structure, resolving in particular its gluon content.

The paper is demonstrating the scientific approach, and is advertising upcoming results from an ongoing measurement campaign by the same authors, aiming to extend its kinematical reach. As such, it is clearly suited for publication in a specialized journal in the particular area of nuclear and hadronic physics (e.g. Phys.Rev.C or Nucl.Phys.A). It does however not meet the criteria of Nature Communications in terms of broad interest and groundbreaking nature of the results, and can thus not be recommended for acceptance.

Reviewer #2 (Remarks to the Author):

This is an interesting paper worthy of publication in Nature Communications. The authors make the first experimental determination of the BH-DVCS interference. Performing fits to their precision data they find that the simplest leading twist and leading order (LT/LO) fit is considerably inferior to including either NLO corrections or twist-3 corrections, thus providing evidence of gluonic degrees of freedom at work in this process. The result will be of interest to QCD physicists working on the structure of the proton and inspire new theory and experimental investigations with the expanded kinematics now available at Jefferson Laboratory.

I have several comments about the presentation that the authors might look at before publication:

1. For expert QCD readers who are not necessarily experts on DVCS and GPDs, I suggest adding a review reference at the end of the opening paragraph "Being a spin 1/2 particle ... for gluons [review(s)]" for definition and more detailed physics explanation of the different GPDs.

2. Just before Eqs.4-6, when the $\{|\mathcal{F}\}$ are written in terms of the new F variables, the text implies that the new F have already been defined in the text whereas they are being defined in Eqs.4-6. This

sentence could be better written. The first time the $F_{\{\lambda\lambda\}}$ appears they have not been defined.

3. Bottom of column 1, page 5: "Two additional fits ... (a) to include genuine twist-3 contributions or (b) ... gluon transversity GPD contributions". It should be stated explicitly after "(b)" that this is the NLO fit. Otherwise the text is confusing.

4. Regarding the authors' fits, I understand from the text that the authors' analysis is purely phenomenological with aim to test for the possible presence of NLO gluon transversity and/or twist 3 GPD effects. The authors do not give details of their best fit parameters in the paper. Since there are theoretical calculations of twist-3 GPDs and gluon transversity it would be helpful if they could comment whether their range of fit parameters are consistent with the size of effects anticipated by these models. Within the present accuracy of experiment and theory detailed comparison at this stage is not necessary. However, it would be useful to know whether the authors' best fit parameters are within order of magnitude of present model expectations before concluding that both the NLO and twist-3 scenarios are equally probable with current data. It is very promising that the forthcoming 12 GeV data from JLab will help resolve this issue from the experimental side.

Reviewer #3 (Remarks to the Author):

This paper presents a first determination of the interference between the DVCS and BH processes by separating the different kinematic dependence of the components. The paper also claims a sensitivity to higher order effects and the gluon density. The analysis is novel and I recommend publication. However I do have some questions and comments to the draft which should be addressed first. In general some more experimental details would help the reader understand the measurements.

- The kinematic range of the measured cross sections in Q^2 , $-t$ and x are somewhat unclear and should be clarified (text below fig 2 on p3, right column)
- in fig 3 show the contribution of the accidentals and the π^0 to motivate the cut in $MX^2 > 0.5$, or alternatively state how large each contribution is. Otherwise the reader cannot see that the signal:bg ratio is improved by this cut.
- how large are the acceptance corrections across all bins? (what range of values does it cover?)
- why do the authors not show the complete set of cross section measurements in all bins? I would recommend publication of the full set of data
- some explanation of the measurement uncertainties is needed. How large are the experimental systematic uncertainties compared to statistical uncertainties of the data? Which systematics are the largest contributions to the measured cross sections? Do the data in fig 4 just show the stat errors? Please clarify.
- how are the measurement systematic uncertainties treated in the fit, and how are the systematics correlated across measurement bins for fixed $-t$?
- how large are the bin migrations in $|t|$ and q^2 ? Some more details should be given explaining how detector resolution effects are treated and corrected for.

- fig 5: state what the fit uncertainty bands represent

- how well do the fits perform when removing the constraints on E_{++} and $E_{\sim ++}$? Presumably there is no significant change in the fit quality, in which case this can be stated. Keeping the number of degrees of freedom fixed in all fits is not a good justification for fixing parameters.

- please add information about the KM15 fit in the body of the text

Dear reviewers,

The Jefferson Lab Hall A Collaboration thanks all three referees for their careful reading and thoughtful comments.

We have made several changes and clarified several points following your recommendations, improving consequently the quality of the manuscript thanks to you.

Please find below a detailed answer, in black, to each of the points or questions you raised.

Finally, we append a detailed list of manuscript revisions.

Sincerely yours,

Maxime Defurne (for the Jefferson Lab Hall A Collaboration)

Reviewer #1 (Remarks to the Author):

The manuscript presents a new measurement of the scattering cross section for the reaction $e+p \rightarrow e+p+\gamma$.

The scattering amplitude for this process receives two types of contributions: photon radiation off the electron (called Bethe-Heitler process, BH) and photon radiation off the proton's constituents (called deeply virtual compton scattering, DVCS). The BH contribution can be predicted in QED, while the DVCS contribution can in principle be used to extract information on the inner structure of the proton. Both reactions interfere in the squared amplitude that determines the cross section.

The measurement presented here improves upon earlier measurements of this scattering reaction in terms of precision and kinematical reconstruction. Moreover, fits to various phenomenological models are provided. These fits, combined with the energy dependence of the cross section, are used subsequently to disentangle the cross section contributions from the square of DVCS and the DVCS-BH interference. This (model dependent) extraction, Fig.5, is the main novel result of this paper.

The Hall A Collaboration thanks you for your clear assessment of our results. We absolutely agree that this paper reports novel results on DVCS improving previous measurements of this reaction.

Although scientifically sound and interesting to its specific subfield as a proof-of-principle on how to disentangle the different contributions to the $e+p \rightarrow e+p+\gamma$ reaction, this result falls very much short off the promise of the paper that is made in the title and abstract: providing a 3D holographic measurement of the proton structure, resolving in particular its gluon content.

We respectfully disagree that the title and abstract oversell the results reported in the paper. While we begin by motivating the goal of this kind of measurements, we do not claim that we will do the holography of the proton's gluon content, something that no single experiment can do. We argue that these results provide evidence of gluonic degrees of freedom at work in this process as pointed out by reviewer#2, something very exciting and never observed before in the valence quark region.

The paper is demonstrating the scientific approach, and is advertising upcoming results from an ongoing measurement campaign by the same authors, aiming to extend its kinematical reach. As such, it is clearly suited for publication in a specialized journal in the particular area of nuclear and hadronic physics (e.g. Phys.Rev.C or Nucl.Phys.A). It does however not meet the criteria of Nature Communications in terms of broad interest and groundbreaking nature of the results, and can thus not be recommended for acceptance.

We have made several changes and clarified the points raised by reviewers #2 and #3. By addressing their specific recommendations, we are confident that the revised version of the manuscript is now more easily accessible to the general reader of Nature Communications.

Reviewer #2 (Remarks to the Author):

This is an interesting paper worthy of publication in Nature Communications. The authors make the first experimental determination of the BH-DVCS interference. Performing fits to their precision data they find that the simplest leading twist and leading order (LT/LO) fit is considerably inferior to including either NLO corrections or twist-3 corrections, thus providing evidence of gluonic degrees of freedom at work in this process. The result will be of interest to QCD physicists working on the structure of the proton and inspire new theory and experimental investigations with the expanded kinematics now available at Jefferson Laboratory.

I have several comments about the presentation that the authors might look at before publication:

1. For expert QCD readers who are not necessarily experts on DVCS and GPDs, I suggest adding a review reference at the end of the opening paragraph "Being a spin 1/2 particle ... for gluons [review(s)] " for definition and more detailed physics explanation of the different GPDs.

Thank you for the suggestion. A reference has been added.

2. Just before Eqs.4-6, when the \mathcal{F} are written in terms of the new F variables, the text implies that the new F have already been defined in the text whereas they are being defined in Eqs.4-6. This sentence could be better written. The first time the $F_{\{\lambda\lambda\}}$ appears they have not been defined.

We have clarified this point by introducing $F_{\{\lambda\lambda\}}$ in a sentence before Eqs. 4-6.

3. Bottom of column 1, page 5: "Two additional fits ...(a) to include genuine twist-3 contributions or (b) ... gluon transversity GPD contributions". It should be stated explicitly after "(b)" that this is the NLO fit. Otherwise the text is confusing.

This is now clarified in the revised version of the manuscript.

4. Regarding the authors' fits, I understand from the text that the authors' analysis is purely phenomenological with aim to test for the possible presence of NLO gluon transversity and/or twist 3 GPD effects. The authors do not give details of their best fit parameters in the paper.

Since there are theoretical calculations of twist-3 GPDs and gluon transversity it would be helpful if they could comment whether their range of fit parameters are consistent with the size of effects anticipated by these models. Within the present accuracy of experiment and theory detailed comparison at this stage is not necessary. However, it would be useful to know whether the authors' best fit parameters are within order of magnitude of present model expectations before concluding that both the NLO and twist-3 scenarios are equally probable with current data. It is very promising that the forthcoming 12 GeV data from JLab will help resolve this issue from the experimental side.

Twist-3 effects are expected to be around ~10% according to available models, which is of the order of the effect we see in the cross sections. While there are existing NLO calculations, these do not include yet any effect of gluon transversity GPDs. The NLO effect currently calculated are included in the leading order Compton Form Factors defined by Braun et al. We have added this information to the text, but we are not able to make a statement about the most probable scenario (higher twist or NLO) at this point.

Reviewer #3 (Remarks to the Author):

This paper presents a first determination of the interference between the DVCS and BH processes by separating the different kinematic dependence of the components. The paper also claims a sensitivity to higher order effects and the gluon density. The analysis is novel and I recommend publication. However I do have some questions and comments to the draft which should be addressed first. In general some more experimental details would help the reader understand the measurements.

- The kinematic range of the measured cross sections in Q^2 , $-t$ and x are somewhat unclear and should be clarified (text below fig 2 on p3, right column)

We have included the kinematics in a table in the revised version of the manuscript, instead of within the text. This should make it more clear.

- in fig 3 show the contribution of the accidentals and the π^0 to motivate the cut in $MX^2 > 0.5$, or alternatively state how large each contribution is. Otherwise the reader cannot see that the signal:bg ratio is improved by this cut.

We have added the requested contributions in Fig. 3. While the effect is already visible now, it is even more significant in bins at large $-t$ and small ϕ , where the relative rate of accidentals is higher than the average shown in Fig. 3. The caption has been modified accordingly to describe the new information included.

- how large are the acceptance corrections across all bins? (what range of values does it cover?)

Geometrically, the acceptance is close to 100% for most of the bins in t and ϕ . However, it significantly drops when getting closer to the edges of the calorimeter which is the case for bins at small ϕ and the largest value of $-t$. For these bins, the acceptance ranges from 30% to 75% depending on the kinematic setting. A sentence has been added to the text to explain this.

- why do the authors not show the complete set of cross section measurements in all bins? I would recommend publication of the full set of data

Showing all data in the paper would require 11 additional figures like Fig. 4. Instead, data points are now available on HEPData website at the URL indicated in the section “Data availability” in the latest version of the article.

- some explanation of the measurement uncertainties is needed. How large are the experimental systematic uncertainties compared to statistical uncertainties of the data? Which systematics are the largest contributions to the measured cross sections? Do the data in fig 4 just show the stat errors? Please clarify.

The systematic uncertainties are described in the 2nd (and end of 1st) paragraph in page 4. Their total value is 3.9% for the unpolarized cross section and 4.4% for the helicity-dependent cross section. The largest contributions are the uncertainties related to the exclusivity cut, the radiative corrections, and the luminosity measurement. The total systematic uncertainties are about the same size as the statistical uncertainties of the unpolarized cross section. We have added a sentence in the revised version that states this. The relative statistical uncertainty on the helicity-dependent cross sections is naturally larger, since the cross section values are much smaller.

We have clarified that error bars in Fig. 4 are only statistical.

- how are the measurement systematic uncertainties treated in the fit, and how are the systematics correlated across measurement bins for fixed $-t$?

In our initial version of the manuscript, only statistical uncertainties were taken into account in the fit. Following the reviewer's question, we have improved the treatment of uncertainties in the following way.

Point-to-point systematic uncertainties have been added quadratically to statistical uncertainties when performing the fit. To account for correlated uncertainties, the fit was performed by shifting all data within each setting by ± 1 sigma. Since there are 2 beam-energy settings in each fit that can have an independent systematic error, this procedure involves 4 fits (resulting in 4 values of χ^2). The effect of the correlated systematic uncertainties was found negligible and only the statistical (+ point-to-point systematics) are reported in the χ^2 values shown in Tab. 2 (of the revised manuscript). This has been clarified in the new version of the paper.

We have also clarified which of the systematics listed in page 4 are point-to-point and which ones are correlated.

- how large are the bin migrations in $|t|$ and q^2 ? Some more details should be given explaining how detector resolution effects are treated and corrected for.

The bin migration is about 10% in t and ϕ . There is no bin migration in q^2 since different measurements in q^2 correspond to different experimental settings. We have added more information on this in the revised version and added a reference where the procedure to account for detector resolution and bin migration effects is described in detail.

- fig 5: state what the fit uncertainty bands represent

Done. Thank you!

- how well do the fits perform when removing the constraints on E_{++} and $E_{\sim++}$? Presumably there is no significant change in the fit quality, in which case this can be stated. Keeping the number of degrees of freedom fixed in all fits is not a good justification for fixing parameters.

Indeed, the fit quality does not change, neither does the separation. The authors agree that keeping the number of degrees of freedom fixed in all fits is not a good justification for fixing parameters. The point the authors are trying to make is that higher twist or NLO contributions are necessary to reproduce the results, without necessarily increasing the number of degrees of freedom in the fit. From all 4 GPDs, E and E_{\sim} are the ones expected to have a smaller contribution to the cross section. We have revised the corresponding part of the paper to address this question.

- please add information about the KM15 fit in the body of the text

Done.

=====

List of revisions:

- 1) Reference [5] added in the introduction.
- 2) Tab. I added in order to better display the kinematics of the measurements. Corresponding text in the main body of the manuscript is now simplified.
- 3) Details of experimental acceptance variations added in page 4 (paragraph 2).
- 4) More details on systematic uncertainties are included now.
- 5) Fig. 3 has been replaced with a more complete version showing more background contributions.
- 6) New CFF $F_{\lambda, \lambda'}$ are now defined in the text.
- 7) Caption of Fig. 4 now describes the uncertainty bars.
- 8) KM15 fit is now explain in the body of the manuscript, and a better reference for it provided [26].
- 9) The procedure to account for systematic uncertainties in the fit is now included.
- 10) More details on the fit parameters and results have been added in page 6.
- 11) A sentence stating the availability of the data from the authors has been added.
- 12) Caption of Fig. 5 now describe the bands shown.

13) Some comment on expectations from theory has been added to the conclusion.

14) To comply with format requirements (checklist) we have:

- Shortened the abstract
- Added section titles and headings
- Added a sentence at the end of the introduction to summarize the results and conclusions
- Formatted the references in Nature Communications style
- Defined error bars as s.d. in the legends
- Labelled figure panels with a single letter
- Added a title to Tab. II

REVIEWERS' COMMENTS:

Reviewer #1 (Remarks to the Author):

This reviewer provided confidential remarks indicating that the publication decision is of editorial nature.

Reviewer #2 (Remarks to the Author):

The authors have included all the requests in my first report and I am now happy to recommend publication of the manuscript in Nature Communications.

This is an interesting paper making the first measuremental determination of BH-DVCS interference. The authors' fits to their data show evidence for gluonic degrees of freedom at work in this process within the authors' kinematics. The result will be of interest to QCD physicists working on the structure of the proton and inspire new theory and experiments at Jefferson Laboratory. The paper will also be of interest to the broader community of physicists who follow fresh developments at the interface of particle and nuclear physics without necessarily being expert in all details.

Reviewer #3 (Remarks to the Author):

Thank you for considering my comments, and those from the other reviewers. I have read your responses and the new manuscript. In general I like the updates you have made to the draft.

The addition of table 1 now clearly explains the kinematic regions used, and the modification to fig 3 is a nice improvement. It is nice to see that the full measurement tables will appear on HepData and that this is now mentioned in the text.

Finally the change in treatment of the systematic uncertainties in the fit leads to a welcome improvement in the χ^2 values, which are quite substantial in some cases as shown in table II. This does not however change any of the conclusions of the paper.

The revised draft is improved and I am happy for it to be published.